# Gender Differences in the Psychopathology of Obesity: How Relevant Is the Role of Binge Eating Behaviors?

**DOI:** 10.3390/brainsci12070955

**Published:** 2022-07-21

**Authors:** Chiara Di Natale, Lorenza Lucidi, Chiara Montemitro, Mauro Pettorruso, Rebecca Collevecchio, Lucia Di Caprio, Luana Giampietro, Liberato Aceto, Giovanni Martinotti, Massimo di Giannantonio

**Affiliations:** 1Department of Neuroscience, Imaging and Clinical Sciences, “G. d’Annunzio” University, 66100 Chieti, Italy; dinatalechiara91@gmail.com (C.D.N.); chiara.montemitro@gmail.com (C.M.); mauro.pettorruso@hotmail.it (M.P.); rebeccacollevecchio@gmail.com (R.C.); giovanni.martinotti@unich.it (G.M.); digiannantoniomassimo@gmail.com (M.d.G.); 2Psychiatry Department, Madonna del Soccorso Hospital, 63074 San Benedetto del Tronto, Italy; dicapriolu@virgilio.it; 3Psychiatry Department, SS. Annunziata Hospital, 66100 Chieti, Italy; luana.giampietro@asl2abruzzo.it; 4Obesity Surgery Center, SS. Annunziata Hospital, 66100 Chieti, Italy; liberato.aceto@asl2abruzzo.it; 5Department of Clinical and Pharmaceutical Sciences, University of Hertfordshire, Hatfield AL10, UK

**Keywords:** obesity, bariatric surgery, gender, psychopathology, binge eating disorder, body image dissatisfaction

## Abstract

Background: Obesity is a condition that affects humans both physically and mentally. Moreover, many psychopathological conditions can be observed in obese patients that may threaten the positive outcomes of bariatric surgery. Purpose: The aim of this study was to identify the main psychopathological correlates of obese candidates for bariatric surgery, with particular attention on the relationship between psychopathology and gender. Methods: In total, 273 candidates for bariatric surgery for obesity underwent a psychiatric evaluation using a compilation of psychometric scales: the Revised Symptom Checklist 90-R (SCL-90-R), the Eating Disorder Examination Questionnaire (EDE-Q), the Binge Eating Scale (BES), the Body Uneasiness Test (BUT) and the Obesity-Related Well-Being (ORWELL 97). The sample was divided on the basis of gender and binge eating disorder (BED) severity. Comparisons between the groups were performed using an analysis of variance model (ANOVA) or a Pearson’s chi-squared test. Further, we also divided our sample into a severe binge eating group (score > 27), a mild to moderate group (18 < score < 26) and a low/no symptoms group (score < 17). Results: Male and female subjects showed different results for the BES, with higher scores reported among women (17.50 ± 9.59) compared to men (14.08 ± 8.64). Women also showed higher scores across most of the SCL-90-R domains and worse outcomes in terms of quality of life. Both women and men in the severe binge eating group reported higher scores for the SCL-90-R. Conclusion: The symptoms of BED, along with body image dissatisfaction (BID), are among the most important to investigate for candidates for bariatric surgery in order to improve the surgery outcomes. Level of evidence: Level III as the evidence came from a cohort analytic study.

## 1. Introduction

Obesity is becoming one of the biggest global health challenges for the coming decades given the progressive increase in its prevalence and its significant consequences on psychophysical health. Obesity is generally defined using the body mass index (BMI), which corresponds to the ratio of body weight to height (kg/m^2^); adults are considered to be overweight when their BMI is between 25–29.9 kg/m^2^ and obese when their BMI is over 30 kg/m^2^ [1].

From an epidemiological point of view, obesity is a global pandemic, with a worldwide presence of about 2.1 billion overweight or obese individuals. It has been estimated that globally 37% of men and 38% of women have a BMI of over 25 kg/m^2^ [2].

Obesity is also the fifth leading cause of death globally. The associated reduction in life expectancy affects both sexes equally but is significantly higher in subjects who are severely obese [3,4,5].

Obesity has significant repercussions on the mental health of affected subjects, which are linked to the impairment of their quality of life and the associated low self-esteem and distortion of body image. Obese subjects are typically discriminated against and have a lower level of education; in the specific case of women, they also have lower economic income and a lower likelihood of marriage [6,7,8]. As for psychiatric disorders, Duarte-Guerra et al. carried out interviews with 393 obese subjects who had been referred to bariatric surgery services. Psychiatric disorders were found in 57.8% of the enrolled subjects, with anxiety and mood disorders being the most common [9]. In the specific case of depressive disorders, their correlation with obesity seems to go both ways: for people with a diagnosis of a depressive disorder and/or those using antidepressant drugs, the risk of developing obesity is higher than for general population; likewise, among adult subjects suffering from depression (especially females), it is very likely to find the condition of being overweight or obese [6,10]. Although this is not a contraindication to surgery, these patients have a high chance of post-surgical depression and lower chances of losing weight to the expected extent [11]. In this sense, recent evidence has suggested the importance of screening for depression and other psychiatric disorders before surgery [12,13] because of the fact that the prevalence of psychopathological conditions among candidates for bariatric surgery is greater than that among the general population [14].

BED is defined by the presence of frequent episodes of uncontrolled excessive food intake with the absence of compensation behavior and it has a significant correlation with overweight- and obesity-related conditions [15,16]. Patients with BED develop obesity with a frequency that is three to six times greater than those without BED and 30% of BED patients were obese children; moreover, the prevalence of BED varies between 3.3 and 5.5% among obese subjects and a familiarity with BED represents a risk factor for the development of obesity in adulthood [15,16,17]. The disorder most commonly affects young adults between 15 and 27 years of age [15] and the female sex. The fact that the disorder has an uneven distribution between the sexes is mainly due to hormonal factors (e.g., estrogen in women) [18].

From an etiopathogenetic point of view, BED is the result of complex interactions between individual biological factors and sociocultural environmental factors. In fact, BED is highly heritable, with an important link to dopaminergic and opioid system genes [16,19]. Psychological factors that are typically associated with BED include low self-esteem, marked dissatisfaction with body image, perfectionism and emotional dysregulation, while diet and nutritional habits may be considered as the most important behavioral risk factors [16]. From a social and cultural standpoint, the most important factors that are involved in the development of BED seem to be the idealization of the concept of beauty and thinness, with women seeming to be more susceptible to this idealization [16,18].

Bariatric surgery represents a therapeutic option for the treatment of obesity-related conditions that are not solvable by non-surgical interventions. According to the criteria that were established by the National Institutes of Health, surgery should be considered for patients with a BMI of over 40 kg/m^2^ and those with a BMI between 35 and 40 who also have associated comorbidities [20]. The only neurological and psychiatric conditions that are considered to be absolute contraindications to surgical intervention include confirmed dementia, severe oligophrenia (IQ < 50) and psychosis, as well as substance use disorders and bipolar disorder when not stabilized. Mood and/or anxiety disorders, eating disorders and personality disorders are considered to be relative contraindications, which are likely to be reassessed after appropriate therapy [21].

The aim of our study was to highlight the psychopathological aspects of obese candidates for bariatric surgery, with particular attention on their correlations with the gender, from the perspective of clustering in order to help to predict post-operative outcomes.

## 2. Materials and Methods

The current study was conducted at the Bariatric Surgery Department of the SS. Annunziata Hospital in Chieti (Italy). The sample consisted of 273 subjects who were over the age of 18 and were diagnosed with obesity, both by the bariatric surgeon who gave the indication for surgery and by a psychiatrist at the Psychiatric Outpatient Clinic of the same hospital. All patients who received the indication for surgery were included in our study, after the provision of their informed consent.

The evaluation consisted of two phases.

Phase 1 (pre-surgery): all participants received a detailed explanation of the study design and were asked to provide informed consent, according to the Helsinki Declaration (1997). The participants underwent a psychiatric examination, a psychometric evaluation and an evaluation of their suitability for surgical treatment.

Phase 2 (post-surgery): patients underwent three follow-up visits at 6 months (T1), 12 months (T2) and 24 months (T3) after surgery. Psychiatric and psychometric evaluations were performed at each timepoint.

At the time of this analysis, the recruitment was suspended because of the COVID-19 pandemic as most non-urgent surgeries were delayed and many of the patients who were enrolled at the baseline were not able to continue with their procedures. For these reasons, the analysis that is presented in the current study only focuses on the baseline measurements that we collected from July 2015 to January 2020.

Psychometric measures. The following baseline measurements were collected in order to provide a complete psychological and psychiatric evaluation of the enrolled subjects.

The Revised Symptom Checklist 90-R (SCL-90-R) [22] is a self-administered questionnaire that aims to investigate the severity of the psychiatric symptoms that were experienced by the patient during the previous week. By adding the scores that correspond to the various items, it was possible to identify the presence of psychopathological aspects, which are highlighted by the following subscales: somatization, obsessive–compulsive, interpersonal sensibility, depression, anxiety, hostility, phobic anxiety, paranoid ideation, psychoticism and additional items.

The Eating Disorder Examination Questionnaire (EDE-Q) [23] is a self-administered questionnaire that aims to evaluate the presence of behaviors that are related to eating disorders in the previous 4 weeks and assign severity and frequency scores to items that are related to the following subscales: restriction, concern for nutrition, concern for body shape and concern for weight.

The Binge Eating Scale (BES) [24] is a self-administered test that evaluates uncontrolled eating disorder in obese subjects. A diagnosis of binge eating disorder is very likely when the overall score is over 27, the presence of some symptoms of binge eating is possible when the score is over 17 and unlikely when the score is less than 17.

The Body Uneasiness Test (BUT) [25] is a self-administered psychometric test that evaluates body image disorders and is divided into BUT A (with the subscales of weight phobia, body image concerns, avoidance, compulsive self-monitoring and depersonalization) and BUT B (with a focus on specific worries about particular body parts or functions). Higher scores indicate greater body uneasiness.

The Obesity-Related Well-Being (ORWELL 97) [26] is a self-administered test that aims to investigate the quality of life of obese patients, particularly in terms of the presence of physical symptoms, the psychological state and social adaptation. Higher scores indicate a lower quality of life.

Statistical Analysis. The demographic data and baseline evaluations are presented as continuous variables with their means and standard deviations and discrete data are summarized using measurements and percentages. The main sample was divided on the basis of gender (male or female) and BED severity (absent, mild to moderate and severe). Comparisons between the groups were performed using ANOVA and Pearson’s chi-squared tests for the continuous and discrete variables, respectively. Potential covariates were evaluated for inclusion on a model-by-model basis, such that covariates that significantly predicted the outcome measures were retained in a linear mixed model. The covariates that were evaluated were gender, age, BMI and the severity of BED. The final linear model, which is presented here, included both gender and BED severity. All statistical tests were two sided and were tested at a 5% level of significance.

## 3. Results

Participant Characteristics. A total of 273 subjects provided their informed consent, completed the psychometric evaluation and were enrolled in the bariatric surgery protocol and were therefore included in the study. The demographic characteristics of the subjects are listed in the Appendix A. As reported previously, the recruitment was completely unbiased and open to all patients who were eligible for bariatric surgery, although the sample was gender-unbalanced with 73 men and 200 women being recruited and successfully enrolled.

Gender-related differences. Given the difference between the numbers of men and women who were included and the potential role of gender bias in the psychological distress that is associated with obesity and being overweight [27], we looked into the two gender groups separately in order to minimize the possible gender bias in our analysis (Appendix A). There were no differences between the two groups in terms of age and years of education, although they differed in terms of occupational status (91.8% of men were employed vs. 46.5% of women; *p* < 0.001) and BMI (M: 44.13 ± 7.36; F: 40.95 ± 6.77; *p* < 0.001). In terms of eating behaviors, the male and female subjects produced different scores for binge eating disorder, with higher scores for BES among the women (17.50 ± 9.59) than among the men (14.08 ± 8.64). At the same time, when we investigated the binge eating disorder severity classes (Marcus, Wing and Hopkins, 1988), we only found small differences between severity class representation in the two groups (Appendix A). As expected, the women produced higher scores across all of the domains that were evaluated by the Body Uneasiness Test and most of the eating disorder evaluation subscales, although no differences were found in terms of restriction (Appendix A).

Regarding psychopathological features, both groups reported scores that were above the general population cut-offs (see below regarding SCL-90-R and psychological distress). The women also showed higher scores across most of the SCL-90-R domains and worse outcomes in terms of their quality of life, as assessed by the ORWELL 97 questionnaire.

Binge Eating Disorder. Given the high impact of binge eating disorder on both bariatric surgery outcomes [21] and psychopathology and psychological distress [28,29,30], we divided our sample into three subsamples on the basis of the Binge Eating Scale: a severe binge eating group, including people with scores that were higher than 27 (n. 50); a mild to moderate group, including people with scores of between 18 and 26 (n. 70); and a group with low/no reported symptoms (with scores that were less than 17) who did not meet the diagnostic criteria for a binge eating disorder (n. 153). The demographic characteristics, psychometric measurements and binge eating-related differences are reported in the Appendix A. The three groups did not differ in terms of gender, age, occupational status, education or BMI. As expected, the three groups differed in terms of all of the domains in both the EDE-Q and the BUT, except for the restriction subscale of the EDE-Q (which was consistent across all groups). At the same time, both the number and severity of mental health symptoms differed between the groups (as reported using the SCL-90-R), with higher scores being reported by people in the severe binge eating group (Appendix A).

SCL-90-R and Psychiatric Symptoms. As mentioned above, the SCL-90-R was administered in order to evaluate a broad range of psychiatric symptoms. In the main sample, the SCL-90-R Global Severity Index varied from a minimum of 0 to a maximum of 3.13 (mean score: 0.79 ± 0.60).

Looking in more detail, we could observe that SCL-90-R mean scores were above the standard cut-offs (Schmitz, Hartkamp and Franke 2000) for psychiatric illness among the general population (Figure 1; Appendix A). In particular, the mean scores for somatization (1.08 ± 0.73), interpersonal sensibility (0.92 ± 0.82), depression (0.91 ± 0.78) and other symptoms (1.01 ± 0.65) were higher than the general population cut-offs (0.9), while the mean scores for obsessive–compulsive (0.87 ± 0.69) and paranoidal ideation (0.80 ± 0.72) were just above the cut-offs (Figure 1).

Moreover, by applying the general population cut-offs, we observed that 32.5% (*n* = 89) of the total sample reported a Global Severity Index score that was higher than that of the general population. Nevertheless, 54% (*n* = 149) of the participants reported mild to severe symptoms for somatization, 40% (*n* = 109) reported mild to severe symptoms for depression, 38.5% (*n* = 105) reported mild to severe symptoms for interpersonal sensitivity, 37.7 % (*n* = 103) reported mild to severe symptoms for obsessive–compulsive and 35.5% (*n* = 97) of the participants reported paranoidal ideation (Figure 2).

The percentage of people who reported scores that were above the cut-offs for the different SCL-90-R domains varied across the binge eating disorder severity classes, as shown in Figure 3.

However, we did not find any effects of BED severity, gender or gender * BED severity interaction on the percentage of people who reported scores that were higher than the general population cut-offs for any of the SCL-90-R domains.

The linear mixed-effects model showed a significant impact of binge eating disorder severity on the Global Severity Index scores for the SCL-90-R, with higher scores in both the severe and mild to moderate binge eating groups compared to the group with no reported BED symptoms (severe: ß = 0.772, *p* = 0.0002; mild to moderate: ß = 0.375, *p* = 0.016). The same significant effect of BED severity was found on the number of positive symptoms that were reported, according to the PST subscale (severe: ß = 27.775, *p* = 0.0002; mild to moderate: ß = 15.338, *p* = 0.0065), and the distress that was associated with those symptoms, according to the PSDI subscale (severe: ß = 2.27, *p* = 0.0339; mild to moderate: ß = 2.025, *p* = 0.0067). There were, however, no significant effects of gender nor any interaction effects between gender and BED severity. A similar effect of BED severity was found on the single domains of the SCL-90-R (Figure 4), with higher single scores in the severe and mild to moderate binge eating groups for the domains that are specified in the caption of the figure.

Moreover, a significant effect of gender was found as the women reported higher scores than the men in the following domains: somatization (ß = 0.258, *p* = 0.0303), obsessive–compulsive (ß = 0.257, *p* = 0.0170), interpersonal sensitivity (ß = 0.315, *p* = 0.0123) and depression (ß = 0.255, *p* = 0.0446). However, there were no interaction effects between gender and BED severity for the SCL-90-R domains, except for other symptoms (additional items), for which we observed a significant interaction effect between gender and BED severity as the women in the severe binge eating group reported lower scores than the men in the same group (ß = −0.605, *p* = 0.0212).

Eating Disorder Examination Questionnaire. As mentioned previously, both the gender and binge eating severity-related groups differed in terms of EDE-Q scores, except for the restriction subscale, which was found to be consistent across all groups. The linear mixed-effects model showed a significant effect of BED severity on the total score of the EDE-Q, with higher scores found in both the severe and mild to moderate binge eating groups compared to the group with no reported BED symptoms (severe: ß = 1.324, *p* = 0.0003; mild to moderate: ß = 0.780, *p* = 0.0041).

Moreover, a significant effect of gender was observed as the women reported higher scores than the men (ß = 0.600, *p* = 0.0003). However, no interaction effects between gender and BED severity were found.

According to our previous observations, there were no significant effects of BED severity or gender on the subscale for restriction, while significant effects of both variables were observed for the other subscales, with higher scores reported among female subjects and the severe and mild to moderate binge eating groups (see the caption for Figure 5 for the scores).

Body Uneasiness Test. As reported in the Appendix A, both the gender and binge eating severity-related groups reported different scores across all of the domains of the Body Uneasiness Test. The linear mixed-effects model showed a significant effect of both BED severity and gender on the total Global Severity Index score for the BUT, with higher scores found in both the severe and mild to moderate binge eating groups (severe: ß = 1.561, *p* = 0.0001; mild to moderate: ß = 0.849, *p* = 0.0035) and among female subjects (ß = 0.937, *p* = 0.00001). No interaction effects between gender and BED severity were found. Regarding the number of positive symptoms that were reported (PST subscale) and the related distress that was caused by those symptoms (PSDI subscale), we found a significant effect of both BED severity and gender (PST: severe: ß = 7.461, *p* = 0.0142; mild to moderate: ß = 5.899, *p* = 0.0103; female: ß = 5.567, *p* = 0.0001; PSDI: severe: ß = 1.146, *p* = 0.0024; female: ß = 0.777, *p* = 0.0001). Moreover, an interaction effect between gender and BED severity was found on the PSDI subscale as the female subjects within the severe binge eating group reported less distress being caused by the symptoms compared to the male subjects in the same group (ß = −0.860, *p* = 0.0369).

Similar significant effects of BED severity and gender were found across the subscales of the BUT as higher single scores were found in the severe and mild to moderate binge eating groups and among the female subjects (see the caption for Figure 6 for the scores).

Across all of the domains, the women in the severe binge eating group reported smaller increases in the scores for the subscales than the men in the same group, even though an interaction effect between gender and BED severity was only found for the weight phobia subscale (ß = −1.023, *p* = 0.0450). This last finding could have been due to the fact the women without binge eating disorder presented higher scores than the men in the same group.

## 4. Discussion

Our study sample mainly consisted of women (70% of the sample), which was consistent with other studies involving people with obesity and BED who were seeking bariatric surgery [31,32]. This higher demand for treatment among women is related to the specific consequences of obesity on women’s health, from the risks that are related to fertility, pregnancy and the menopause to those that are related to tumors (especially dermatological) [33]. Moreover, women are also affected by higher levels of stigma for being overweight [34] and they suffer more than men with the same BMI in terms of quality of life and severe psychosocial deficits [27]. According to our evidence, the women in our study reported having a lower quality of life compared to males with the same BMI. Further, the psychopathological load seemed to be higher among women, even though this effect tends to disappear among subjects who have BED. The women were also found to be less comfortable with their body image and more prone to report body image distortions than the men. However, the gender differences that were highlighted in all of the investigated domains were found to disappear among subjects who had a diagnosis of severe BED.

Our results showed that patients with severe BED could experience difficult relationships with their body image and that the severity of the BED was the main factor that drove their body image perception. It is well known that patients with BED have great difficulty in accepting their body and attribute an improper value to weight and body shape. Moreover, BID is associated with dysfunctional eating behaviors, as well as psychiatric comorbidities [16]. Body image is one of the aspects of a person’s mental representation of themselves and is defined as a person’s psychological experience of the appearance and function of their body [35], especially with regard to physical and external characteristics [36]. In this sense, a person’s relationship with their body represents an essential element in the construction of their identity, which conveys personality, social, cultural, psychological and biological factors [37]. Body image travels along a gradient of satisfaction and moving away from the “ideal” image that is produced by a person’s expectations and those of others can lead to the development of real disorders [38]. It is now clear that BID represents one of the main risk factors and an indication of the fallout from conditions such as obesity, various eating disorders [39] and a whole series of psychopathological conditions [35,40]. In the specific case of the BID–obesity relationship, a previous study showed that obese people are subjected to an unacceptable level of stigmatization that is linked to their appearance, which (in most cases) causes them to suffer from the burden of living in their own body and feelings of sadness, shame, frustration and rejection until they renounce having relationships and regularly attending school and work environments [37]. Moreover, BID has been found to be significantly related to low self-esteem and a poor quality of life among subjects with BED, regardless of BMI [41]. Our results confirmed these findings.

On the other hand, our results showed that higher BED severity scores were related to worse scores for all of the subscales of the SCL-90-R. In accordance with the recent literature, these results suggested that patients with BED were at a greater risk of developing psychiatric disorders than subjects without a BED diagnosis [15]. Moreover, a BED diagnosis for obese subjects is so frequently associated with the presence of other psychiatric disorders that it can almost be considered as a marker for psychopathological conditions [14]. The most frequently conditions are anxiety and mood disorders, borderline, obsessive–compulsive and avoidant personality disorders and substance use disorders. Due to these comorbidities, patients with BED may be considered to be at a greater risk of suicide than the general population, especially in the presence of depressive symptoms [16].

Our work highlights the importance of thoroughly screening for the presence of BED in candidates for bariatric surgery. In our sample, this disorder was so pervasive that is overcame gender differences and influenced all psychopathological dimensions.

Furthermore, our study was conducted on a particularly large population, in which we investigated several dimensions of eating behaviors and psychopathology.

However, the fact that 70% of this population was female represented the main limitation of our work and resulted in a gender bias, which has been common among other studies involving obese people who were seeking bariatric surgery.

A further limitation of our study was the lack of post-surgery data due to the COVID-19 pandemic. In fact, the pandemic stopped bariatric surgery for a certain period of time as it is a form of elective surgery [42]. After the first peaks of the pandemic, experts worked toward a cautious and effective resumption of bariatric interventions on the basis of criteria that could be shared as much as possible [43]. Following the outbreak of the pandemic, the incidence of eating disorders increased throughout 2020 after an initial decrease in the early parts of the year (which reflected the reduction in all diagnoses) [44]. The lockdown periods had negative effects on individuals and increased the incidence of stress, anxiety and altered eating habits, which (in some cases) led to eating disorders. This increase especially affected the number of reported BED cases, with a 15.3% increase in the incidence rate [45]. An interesting Italian study found a correlation between the state of anxiety and fear of contagion and pathological eating behaviors, which highlighted the importance of psycho-educational interventions for preventing the risk factors of mental health disorders [46]. The COVID-19 pandemic and obesity can influence each other since obesity represents a risk factor for severe forms of the coronavirus disease [47] and, on the other hand, a review and meta-analysis reported that 65% of individuals with eating disorders experienced worse symptoms during the lockdowns and more than 50% of individuals with obesity reported increased weight, snacking and reduced physical activity [48].

## 5. Conclusions

Because BID and BED play central roles in the psychopathology of obese subjects, it is crucial to evaluate patients who turn to bariatric surgery in order to identify factors that can be associated with negative outcomes and poor adherence to post-operative indications.

Obese patients who seek bariatric surgery to lose weight are the most psychologically affected and have high levels of concern and dissatisfaction in relation to body shape, which negatively correlate with the success of weight-loss therapies [36].

The use of bariatric surgery usually occurs after other dietary, nutritional, physical and pharmacological measures have failed because of the fact that psychopathological discomfort often leads obese patients to choose surgery as the last resort. For this reason, it is important for future studies to better identify and manage psychiatric comorbidities that are related to obesity in order to improve post-surgery outcomes.

## Figures and Tables

**Figure 1 brainsci-12-00955-f001:**
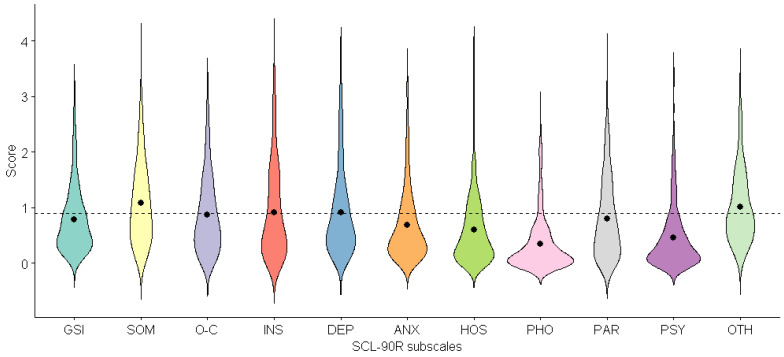
The SCL-90-R scores for obese patients who were eligible for bariatric surgery (distribution and mean values): GSI, Global Severity Index; SOM, somatization; O-C, obsessive–compulsive; INS, interpersonal sensibility; DEP, depression; ANX, anxiety; HOS, hostility; PHO, phobic anxiety; PAR, paranoidal ideation; PSY, psychoticism; OTH, other symptoms (additional items).

**Figure 2 brainsci-12-00955-f002:**
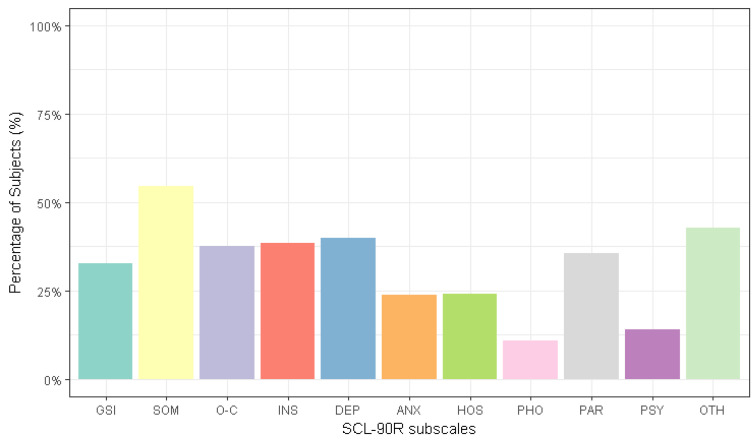
The percentage of subjects with scores that were above the general population cut-offs for the SCL-90-R (total sample): GSI, Global Severity Index; SOM, somatization; O-C, obsessive–compulsive; INS, interpersonal sensibility; DEP, depression; ANX, anxiety; HOS, hostility; PHO, phobic anxiety; PAR, paranoidal ideation; PSY, psychoticism; OTH, other symptoms (additional items).

**Figure 3 brainsci-12-00955-f003:**
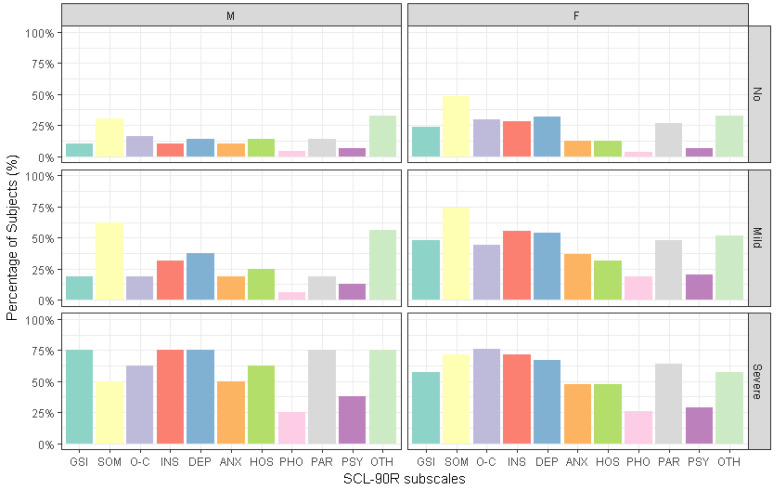
The percentage of subjects with scores that were above the SCL-90-R general population cut-offs, according to binge eating disorder (BED) severity and gender: GSI, Global Severity Index; SOM, somatization; O-C, obsessive–compulsive; INS, interpersonal sensibility; DEP, depression; ANX, anxiety; HOS, hostility; PHO, phobic anxiety; PAR, paranoidal ideation; PSY, psychoticism; OTH, other symptoms (additional items).

**Figure 4 brainsci-12-00955-f004:**
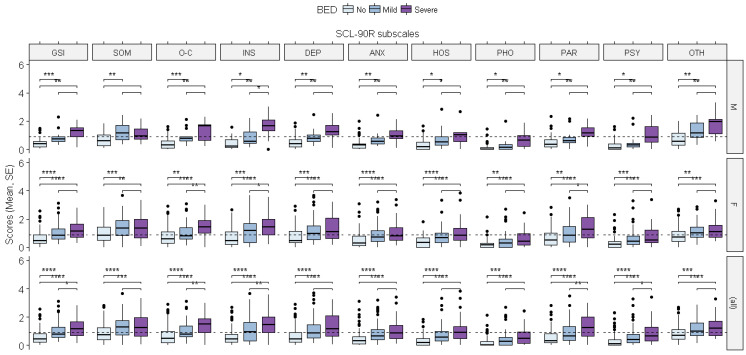
The effects of binge eating disorder severity and gender on SCL-90-R scores. The differences between the groups were evaluated using the Wilcoxon test: dashed lines, general population cut-offs (0.9); GSI, Global Severity Index; SOM, somatization (severe: ß = 0.45, *p* = 0.013; mild to moderate: ß = 0.530, *p* = 0.0076); O-C, obsessive–compulsive (severe: ß = 0.859, *p* = 0.0003; mild to moderate: ß = 0.428, *p* = 0.0169); INS, interpersonal sensibility (severe: ß = 1.249, *p* = 0.00001; mild to moderate: ß = 0.392, *p* = 0.0601); DEP, depression (severe: ß = 0.834, *p* = 0.0025; mild to moderate: ß = 0.406, *p* = 0.0501); ANX, anxiety (severe: ß = 0.704, *p* = 0.0027; mild to moderate: ß = 0.304, *p* = 0.0843); HOS, hostility (severe: ß = 0.620, *p* = 0.0070; mild to moderate: ß = 0.368, *p* = 0.0334); PHO, phobic anxiety (severe: ß = 0.563, *p* = 0.0022); PAR, paranoidal ideation (severe: ß = 0.733, *p* = 0.0038); PSY, psychoticism (severe: ß = 0.788, *p* = 0.0002); OTH, other symptoms (additional items) (severe: ß = 0.969, *p* = 0.0001; mild to moderate: ß = 0.522, *p* = 0.0036). * = *p* < 0.05; ** = *p* < 0.01; *** = *p* < 0.001; **** = *p* < 0.0001.

**Figure 5 brainsci-12-00955-f005:**
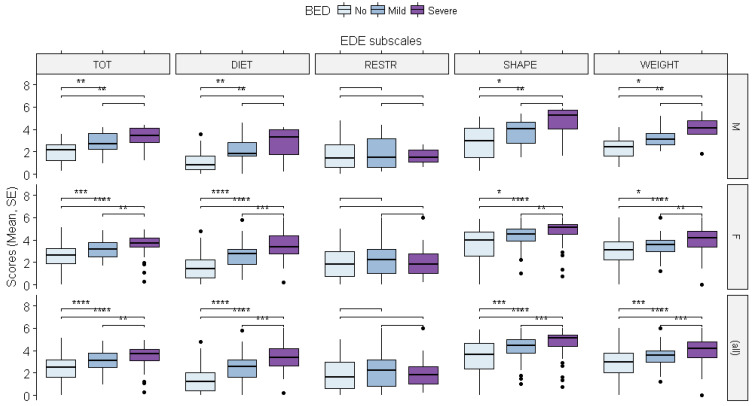
The effects of binge eating disorder severity and gender on the scores for the Eating Disorder Examination Questionnaire (EDE-Q). The differences between the groups were evaluated using the Wilcoxon test: weight-related concerns (severe: ß = 1.714, *p* = 0.0001; mild to moderate: ß = 0.764, *p* = 0.0183; female: ß = 0.741, *p* = 0.0002); body shape-related concerns (severe: ß = 1.900, *p* = 0.0002; mild to moderate: ß = 0.947, *p* = 0.0121; female: ß = 0.959, *p* = 0.00001); diet-related concerns (severe: ß = 1.756, *p* = 0.0001; mild to moderate: ß = 1.106, *p* = 0.0007; female: ß = 0.478, *p* = 0.0145). * = *p* < 0.05; ** = *p* < 0.01; *** = *p* < 0.001; **** = *p* < 0.000.

**Figure 6 brainsci-12-00955-f006:**
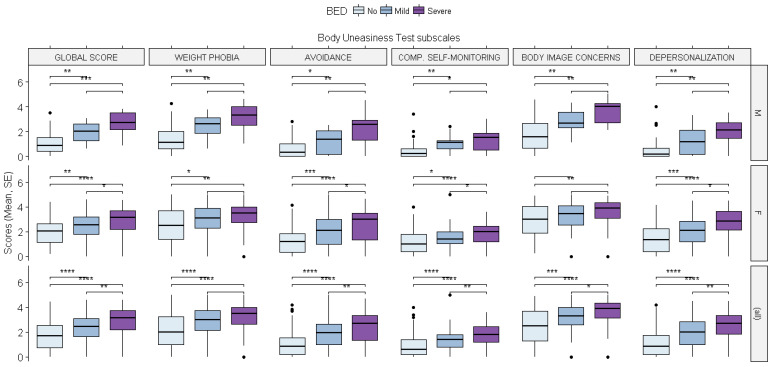
The effects of binge eating disorder severity and gender on the scores for the Body Uneasiness Test. The differences between the groups were evaluated using the Wilcoxon test: avoidance (severe: ß = 1.624, *p* = 0.0003; female: ß = 0.762, *p* = 0.0002); body image concerns (severe: ß = 1.854., *p* = 0.0001; mild to moderate: ß = 1.022, *p* = 0.0032; female: ß = 1.081, *p* = 0.00001); compulsive self-monitoring (severe: ß = 0.909, *p* = 0.0084; mild to moderate: ß = 0.572, *p* = 0.0277; female: ß = 0.694, *p* = 0.00001); depersonalization (severe: ß = 1.384, *p* = 0.0011; mild to moderate: ß = 0.843, *p* = 0.0084; female: ß = 0.902, *p* = 0.00001); weight phobia (severe: ß = 1.724, *p* = 0.0002; mild to moderate: ß = 0.995, *p* = 0.0042; female: ß = 1.083, *p* = 0.00001). * = *p* < 0.05; ** = *p* < 0.01; *** = *p* < 0.001; **** = *p* < 0.000.

## Data Availability

Data presented in this study are available on request from the corresponding author.

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
