# Peer review of "Gender Differences in the Psychopathology of Obesity: How Relevant Is the Role of Binge Eating Behaviors?"

_brainsci, 2022, doi:10.3390/brainsci12070955_

Round 1

Reviewer 1 Report

The authors of the manuscript titled „Gender differences in the psychopathology of obesity: how relevant is the role of binge eating behaviors?” chose a very important and timely topic for their study especially because there have been numerous reports in literature regarding an increase in obesity rates and binge-eating behaviors induced by mental health issues as they relate to COVID-19 pandemic, isolation, staying at home, uncertainty of the future, job instability, financial struggles and more. The text is interesting and it is visible that a lot of work has been put in by the authors to collect and present all the data, however there are some issues with the paper that need to be addressed.

First of all there are no Line markings in the text which makes it very hard to reference particular parts of the manuscript. Each publication has to have all lines of the text numbered – until this issue is fixed, it is hard for me to point out for example some small mistakes that I have noticed as far as the English language goes – although the use of English considering it being the authors’ second language is generally very good.

Secondly, there is a huge matter of all the diagrams, tables and graphs included in the paper. They are placed at the very end of the text which makes it rather inconvenient for the potential reader to quickly access the data collected in them. The should either be placed in sections in which they are mentioned or in the appendix/supplementary file. The tables and graphs are not very readable and clear – they need to be simplified for the purpose of this article, there is also no need to repeat all the numerical data in the text itself, it would be very beneficial to the overall reception of this work to condense these numbers and include only the most important ones in the text and leave the rest for the diagrams.

One last problem is that the literature needs to be urgently updated. There are no works/articles from the year 2022 or 2021, only one from 2020, some from 2019 but most are 5, 10 or even more years older. Taking into consideration that the paper talks about a topic where there are constant changes in the statistics that need to be constantly updated, this has to be corrected and more recent publications have to be included in the „References” section.

Reviewer 2 Report

Overall

-Pay attention to text formatting, as sometimes there is inappropriate font size (for example in the Abstract) or no spaces between the dot ending the sentence and consecutive capital letter.

-There is no need to put the title in the quotation marks.

Abstract

-A short sentence presenting the background for this study is needed to familiarize readers with the topic.

-Please adhere to the same grammar tense while presenting the Abstract, namely “comparisons between the groups is performed with an analysis of variance model” should be changed into “was performed”.

Introduction

-Authors must carefully read the Journal’s guidelines about way of citing articles in the text. Specifically, you should put only the number of the cited article in the square bracket, instead of writing surnames of the manuscripts Authors.

-Paragraph one – you should provide a source of this information. Moreover, I would suggest to merge two first paragraphs into one.

Materials and Methods

-Authors should provide a short information about study sample. What was the sampling of the study participants? Did you perform the calculation of a sample size? You briefly should clarify it.

-What were the inclusion and exclusion criteria for this study?

-Information regarding applied questionnaires is very scarce. For each questionnaire you should add information what is the maximum possible score to obtain (this is necessary for understanding the results in the tables) and provide general information about the number of items within the questionnaire and assesses aspects. Are there any ranges which can be distinguished based on obtained number of points?

Results

-I suggest changing the way of presenting the results. In my opinion, each figure/table should go directly after the description of this figure/table. I think it may facilitate the understanding of the results.

-Tables in their current form do not stick to the Journal’s guidelines. Font and font size is inappropriate and unfortunately some rows in Table 1 are not visible at all.  

-Figure 1 – on the horizontal axis you use some abbreviations, however, they are not explained.

-Figure 4 – why the description of the figure stands next to it, not under the figure?

-The last table is not numbered – what does it present?  

Discussion

-“This higher demand for treatment among women is related to the specific consequences of obesity on women’s health” – what consequences are you talking about?

-Authors should highlight strengths and limitations of this study.

Conclusion

-This part is missing. Every study should contain conclusion which summarizes the whole study and addresses the implications for further research or actions.

Round 2

Reviewer 1 Report

The authors have improved some aspects of the manuscript e.g. correcting the diagrams to be more readable or numbering each line of the text, however the last request to update the literature has not been addressed. Most recent papers, preferably from 2022-2020, need to be added to the references section urgently. Also, now that I can reference each line, there are quite a few examples of errors in the use of English:

Line 10 "affects humanS both physically and mentally" - the use of "dimensions" is incorrect in this context  Line 11 "conditions can be..." - "it" is unnecessary and also "traced" is not the best choice of a word here, I would change it to e.g. observed/witnessed  Line 14 "them and gender" - please specify the "them" and improve the sentence to be more grammatically correct Line 368 "population was composed OF women" - "by" is used only when speaking about composing musical pieces   Line 377 "factor for severE forms"  Please note that these are only some mistakes that I was able to point out but there are quite a few similar, small errors, throughout the whole text. It would be beneficial for the manuscript to be checked by a native speaker or a professional editor before final publication. The manuscript needs major revisions still, in my opinion.

Author Response

The authors have improved some aspects of the manuscript e.g. correcting the diagrams to be more readable or numbering each line of the text, however the last request to update the literature has not been addressed. Most recent papers, preferably from 2022-2020, need to be added to the references section urgently. Also, now that I can reference each line, there are quite a few examples of errors in the use of English: Line 10 "affects humanS both physically and mentally" - the use of "dimensions" is incorrect in this context  Line 11 "conditions can be..." - "it" is unnecessary and also "traced" is not the best choice of a word here, I would change it to e.g. observed/witnessed  Line 14 "them and gender" - please specify the "them" and improve the sentence to be more grammatically correct Line 368 "population was composed OF women" - "by" is used only when speaking about composing musical pieces   Line 377 "factor for severE forms"  Please note that these are only some mistakes that I was able to point out but there are quite a few similar, small errors, throughout the whole text. It would be beneficial for the manuscript to be checked by a native speaker or a professional editor before final publication. The manuscript needs major revisions still, in my opinion.

Thank you very much for allowing us to further improve our paper. As you indicated, we have added more recent literature data about eating disorders referring to the COVID-19 pandemic period. Regarding grammar and use of english, we have tried to make the required improvements. We are waiting for information if further linguistic revision is necessary.

Reviewer 2 Report

Authors have responded to all my suggestions and remarks.

Author Response

We appreciate having satisfied your requests and we thank you for allowing us, with your suggestions, to have made important improvements to our work.